# Mutational Landscape of Alzheimer’s Disease and Frontotemporal Dementia: Regional Variances in Northern, Central, and Southern Italy

**DOI:** 10.3390/ijms25137035

**Published:** 2024-06-27

**Authors:** Claudia Saraceno, Lorenzo Pagano, Valentina Laganà, Andrea Geviti, Silvia Bagnoli, Assunta Ingannato, Salvatore Mazzeo, Antonio Longobardi, Silvia Fostinelli, Sonia Bellini, Alberto Montesanto, Giuliano Binetti, Raffaele Maletta, Benedetta Nacmias, Roberta Ghidoni

**Affiliations:** 1Molecular Markers Laboratory, IRCCS Istituto Centro San Giovanni di Dio Fatebenefratelli, 25125 Brescia, Italy; csaraceno@fatebenefratelli.eu (C.S.); lpagano@fatebenefratelli.eu (L.P.); alongobardi@fatebenefratelli.eu (A.L.); sbellini@fatebenefratelli.eu (S.B.); 2Department of Primary Care, Regional Neurogenetic Centre (CRN), ASP Catanzaro, 88046 Lamezia Terme, Italy; valelagana@gmail.com (V.L.); raffaelegiovanni.maletta@asp.cz.it (R.M.); 3Service of Statistics, IRCCS Istituto Centro San Giovanni di Dio Fatebenefratelli, 25125 Brescia, Italy; ageviti@fatebenefratelli.eu; 4Department of Neuroscience, Psychology, Drug Research and Child Health, University of Florence, 50139 Florence, Italy; silvia.bagnoli@unifi.it (S.B.); assunta.ingannato@unifi.it (A.I.); benedetta.nacmias@unifi.it (B.N.); 5Vita-Salute San Raffaele University, 20132 Milan, Italy; mazzeo.salvatore@hsr.it; 6IRCCS Policlinico San Donato, 20097 San Donato Milanese, Italy; 7MAC-Memory Clinic and Molecular Markers, IRCCS Istituto Centro San Giovanni di Dio Fatebenefratelli, 25125 Brescia, Italy; sfostinelli@fatebenefratelli.eu (S.F.); gbinetti@fatebenefratelli.eu (G.B.); 8Department of Biology, Ecology and Earth Sciences, University of Calabria, 87036 Rende, Italy; alberto.montesanto@unical.it; 9IRCCS Fondazione Don Carlo Gnocchi, 50143 Florence, Italy

**Keywords:** Alzheimer’s Disease, Frontotemporal Dementia, gene, mutation, *APP*, *PSEN1*, *PSEN2*, *MAPT*, *GRN*, *C9orf72*

## Abstract

Alzheimer’s Disease (AD) and Frontotemporal Dementia (FTD) are the two major neurodegenerative diseases with distinct clinical and neuropathological profiles. The aim of this report is to conduct a population-based investigation in well-characterized *APP*, *PSEN1*, *PSEN2*, *MAPT*, *GRN*, and *C9orf72* mutation carriers/pedigrees from the north, the center, and the south of Italy. We retrospectively analyzed the data of 467 Italian individuals. We identified 21 different *GRN* mutations, 20 *PSEN1*, 11 *MAPT*, 9 *PSEN2*, and 4 *APP*. Moreover, we observed geographical variability in mutation frequencies by looking at each cohort of participants, and we observed a significant difference in age at onset among the genetic groups. Our study provides evidence that age at onset is influenced by the genetic group. Further work in identifying both genetic and environmental factors that modify the phenotypes in all groups is needed. Our study reveals Italian regional differences among the most relevant AD/FTD causative genes and emphasizes how the collaborative studies in rare diseases can provide new insights to expand knowledge on genetic/epigenetic modulators of age at onset.

## 1. Introduction

Alzheimer’s Disease (AD) and Frontotemporal Dementia (FTD) are the two major neurodegenerative diseases with distinct clinical and neuropathological profiles that ultimately result in dementia, characterized by substantial synaptic and neuronal loss, leading to brain atrophy [1].

AD is the most common neurodegenerative cause of dementia in the elderly, and currently affects around 50 million patients worldwide [2,3,4]. It is characterized by progressive decline in cognitive domains, encompassing memory loss, behavioral changes, and loss of functional abilities [5,6]. The modification in the brain that causes these alterations is thought to begin many years before the symptoms’ onset [7,8,9,10]. Although the exact etiology of AD remains unknown, two components have been identified thus far as key players in the disease: amyloid-β (Aβ) plaques, formed by the aggregation of intra- and extra-cellular Aβ, and intracellular neurofibrillary tangles (NFT) composed of hyperphosphorylated tau protein accumulation [11,12,13,14]. The majority of AD appears to be sporadic, with patients who exhibit Late-Onset AD (LOAD), defined as AD with an onset later than 65 years [15], while a small percentage of all AD cases are linked to rare and highly penetrant mutations in one of three principal genes: amyloid precursor protein (*APP*) [16,17,18,19], presenilin 1 (*PSEN1*) [20,21], and presenilin 2 (*PSEN2*) [22,23]. Inherited in an autosomal dominant mode, these mutations are linked to an Early-Onset AD (EOAD) before the age of 65, and might cause an alteration of Aβ production, leading to the apoptosis of the neurons and dementia [24,25,26,27]. *APP* encodes for a protein called amyloid precursor protein, whose cleavage by the subsequent action of two enzymes, β- and γ-secretase, leads to the production of the neurotoxic fragment Aβ 1-42. *PSEN1* and *PSEN2*, encoding for presenilin-1 and presenilin-2, are the catalytic subunits of γ-secretase [28,29].

FTD is the second most common form of early-onset dementia, with clinical presentations in individuals under 65 years old [30,31]. It involves the degeneration of the frontal and temporal brain regions and it is marked by abnormalities in personality, language, and executive function [32,33]. FTD encompasses different phenotypes, namely the behavioral variant of FTD (bvFTD) and the agrammatic or the semantic variant of primary progressive aphasia (avPPA, svPPA, respectively) [34,35]. Protein aggregation, glia hyperproliferation and inflammation, lysosomal alteration, and neuronal death are the primary pathogenic features of FTD. Specifically, the most prevalent neuropathological hallmarks are intracellular ubiquitin, TAR DNA-binding protein (TDP)-43-positive inclusions, microtubule-associated protein tau (MAPT), and fused in sarcoma (FUS) protein deposition, present in both hereditary and sporadic FTD [33,36].

In general, up to 40% of FTD patients report a family history of dementia, although only 10% show an autosomal dominant trait [37,38,39]. The most common causative genes are: *MAPT* [40,41], granulin (*GRN)* [42,43], and the chromosome 9 open reading frame 72 (*C9orf72*) [44,45]. *MAPT*, encoding for tau protein, was the first gene found to have a role in families affected by FTD. Mutations in *MAPT* alter the physiological balance of the tau isoform, increasing or decreasing its interactions with microtubules and consequently altering the microtubules’ structural stability [40,46]. *GRN* mutations are the most frequent genetic determinant of familial dementia in Northern Italy [47], and account for around 5% of all FTD cases and up to 25% of familial ones [48]. A decrease in circulating progranulin protein is accounted for by the majority of *GRN* stop codons mutations. Also, alterations in the secretion or processing of progranulin is caused by some missense mutations, resulting in a reduced protein functionality [49]. A total of 25% of familial FTD [36,44,50,51,52] and 6% of sporadic cases [53] are associated with a pathological expansion of the hexanucleotide GGGGCC (>30) in the first intron/promoter of *C9orf72* [36,44,50,51,52,53]. Alleles with up to 25 repeats have been associated with a normal phenotype in a healthy Italian population [54]. The intermediate expansion (12–30) has a risk effect in familial/sporadic FTD, and its repeat unit number influences *C9orf72* expression and disease phenotype in terms of age at onset and associated clinical subtype [44,52,55]. 

Mutations in *APP*, *PSEN1*, *PSEN2*, *MAPT*, *GRN*, and *C9orf72* are well described and updated in countless gene mutation databases. The most specific for AD and FTD are Alzforum (*https://www.alzforum.org/mutations*, accessed on 24 June 2024) and The AD&FTD Mutation Database (*www.molgen.ua.ac.be/ADMutations* or *https://uantwerpen.vib.be/mutations*, accessed on 24 June 2024). All these mutations share autosomal dominant inheritance in familial cases with early onset dementia, and although Mendelian forms represent a small fraction of occurrences, studies conducted on the implicated genes can reveal the underlying mechanisms of these disorders. Broad phenotypic expression variability between or within pedigrees bearing the same mutation characterizes many monogenic conditions. Evidence from cohort studies and individual case series suggested that the age at onset, age at death, and disease duration are highly variable across the genes implicated in FTD, in particular in *GRN*/*C9orf72* pedigrees [56]. Age-related penetrance was described in individuals with *GRN* and *C9orf72* mutations, with *MAPT* mutations usually being fully penetrant. Missense mutations in the *PSEN2* gene may show incomplete penetrance [57], as also reported in a pair of mutated monozygotic twins [58]. Thus, the identification of a mutation is not a certain predictor of disease or age at onset. Substantial variation remains within many AD/FTD families and mutation types, suggesting the existence of genetic or environmental modifiers, both of which could act through epigenetic changes, such as DNA methylation at specific CpG sites [57,58,59,60,61,62,63]. The purpose of the present study, which is based on the collaboration of researchers from the north, the center, and the south of Italy as part of the GARDENIA Consortium, is to conduct a population-based investigation in well-characterized cohorts in Italy, a country with a rich history of cultural influences.

Therefore, in this Italian retrospective cohort study, we aimed to analyze the phenotypic characteristics of the main forms of genetic Alzheimer’s Disease and Frontotemporal Dementia, including age at onset, as well as examining the effect of mutation type (*APP*, *PSEN1*, *PSEN2*, *MAPT*, *GRN*, and *C9orf72* genes). 

## 2. Results and Discussion

Our combined dataset comprised a total of 467 individuals, 349 patients and 118 asymptomatic subjects from 218 pedigrees who had data available for age at onset, sex, mutation, and clinical diagnosis (Table 1): a total of 144 individuals with *GRN* mutations (from 79 families), 125 individuals with *PSEN1* mutations (from 32 families), 79 individuals with *C9orf72* expansions (from 57 families), 58 individuals with *APP* mutations (from 21 families), 34 individuals with *PSEN2* mutations (from 19 families), and 29 individuals with *MAPT* mutations (from 16 families). Interestingly, two individuals carried a double mutation, in *PSEN2/MAPT* and *PSEN2/GRN*, respectively, and a subject carried two different mutations in the *GRN* gene.

In total, 21 different *GRN* mutations, 20 *PSEN1*, 11 *MAPT*, 9 *PSEN2*, and 4 *APP* mutations, were described. The most common mutations are as follows: *GRN* gene, Leu271LeufsX10 (rs63749877; 105 individuals [104 from Northern cohort, 1 from Central cohort], 47 families); *PSEN1* gene, Met146Leu (rs63750306; 58 individuals [52 from Southern cohort, 4 from Northern cohort, and 2 from Central cohort] 3 families); *APP* gene, Ala713Thr (rs63750066; 36 individuals [35 from Southern cohort, 1 from Central cohort], 14 families); *PSEN2* gene, Met239Val (rs28936379; 10 individuals, [9 from Central cohort, 1 from Southern cohort] 3 families); *MAPT* gene, Pro301Leu (rs63751273; 9 individuals [5 from Northern cohort, 4 from Southern cohort], 3 families).

Overall, the most prevalent genetic group was that comprising *GRN* mutation carriers (144 [30.8%] of 467 individuals), followed by *PSEN1* mutation carriers (125 [26.8%]), followed by individuals carrying *C9orf72* expansion (79 [16.9%]), *APP* (58 [12.4%]), *PSEN2* (34 [7.3%]), followed by the least common group with mutations in *MAPT* (29 [6.20%]) (Figure 1a). Moreover, we observed geographical variability in mutation frequencies by looking at each cohort of participants (Figure 1b–d).

In the **Northern cohort** we found a total of 169 subjects from 82 families: 115 with *GRN* mutations (from 55 families [67.1%] of 82), 27 with *C9orf72* expansion (15 families [18.3%]), 10 with *MAPT* mutations (5 families [6.1%]), 9 with *PSEN1* mutations (4 families [4.9%]), 7 with *PSEN2* (2 families [2.4%]), and 1 with *APP* mutation (1 family [1.2%]). In *GRN*, a total of 10 mutations were identified and the most common mutation is Leu271LeufsX10 (104 individuals from 46 families). The Leu271LeufsX10 mutation in exon 7 of *GRN* was first described in Northern cohorts [64,65], and is one of the most common *GRN* mutations worldwide [48]. The Leu271LeufsX10 mutation in exon 7 of *GRN* was then identified in a number of families belonging to the north of Italy, in particular the Lombardy region, suggesting a founder effect from a common ancestor. Performing a haplotype sharing analysis (on 32 families, residents of Lombardy), we previously demonstrated that almost all families can be traced to a single founder; moreover, we estimated the age of this mutation using different methods and population growth rates, both for Italy and Lombardy, and we dated the origin of this mutation to the Middle Ages, at the turn of the first millennium [66]. In *MAPT* we observed 4 mutations, with Pro301Leu as the most frequent (5 individuals from 2 families). For *PSEN1* and *PSEN2* we identified 3 mutations each, with Met146Leu (4 individuals in the same family) and Met239Ile (rs63749884; 4 individuals in the same family) as the most represented, respectively. The only mutation identified for *APP* is Thr719Pro (rs2146237857, 1 individual from 1 family). Interestingly, in this cohort, a patient affected by FTD is a carrier of two distinct mutations in *GRN* gene, Leu271LeufsX10 and Ala505Gly (rs780159686).

In the **Central cohort** we found a total of 112 subjects from 58 families: 50 with *PSEN1* mutations (from 20 families [34.5%] of 58), 29 with *C9orf72* expansion (27 families [46.6%]), 21 with *APP* mutations (6 families [10.3%]), 9 with a mutation in *PSEN2* (2 families [3.4%]), 2 with *MAPT* mutations (2 families [3.4%]), and 1 with a *GRN* mutation (1 family [1.7%]). For *PSEN1* we identified 12 mutations with Cys92Ser as the most frequent (rs63751141; 14 individuals, 5 families), and 3 mutations in *APP* with Val717Ile as the most represented (rs63750264; 19 individuals, 3 families). Met239Val was the only mutation identified in *PSEN2*. For *GRN*, the mutation identified was Leu271LeufsX10, and the mutations identified for *MAPT* were Val755Ile and Ser712Phe (rs63750869; rs63750635).

In the **Southern cohort** we found a total of 186 subjects from 84 families: 66 with *PSEN1* mutations (from 8 families [12.5%] of 84), 36 with *APP* mutations (14 families [16.7%]), 28 with *GRN* mutations (23 families [27.4%]), 23 with *C9orf72* expansion (15 families [17.9%]), 18 with *PSEN2* mutations (15 families [17.9%]), and 17 with *MAPT* mutations (9 families [10.7%]). We found 7 mutations in *PSEN1*, with the pathogenic mutation Met146Leu carried by 52 individuals belonging to the same family (“N family” already described in [67], 2 mutations in *APP* with Ala713Thr as the most frequent (35 individuals, 13 families), 12 mutations for *GRN* with Thr382fs as the most common (rs63750805; 8 individuals, 5 families), 7 mutations for *PSEN2* with the most frequent being Arg62His (rs58973334; 7 individuals, 6 families), and 9 mutations in *MAPT* with Pro301Leu as the most represented. The Met146Leu in *PSEN1* gene was considered as a private mutation, with a founder in the Calabrian population, dated around the year 1000 [68] and shared among several AD patients dispersed across centuries and continents due to emigration flow [67,68]. Interestingly, in the Southern cohort, a subject carried a double mutation in *PSEN2* (Arg62His)-*MAPT* (Gly335Ser, rs63750095). Moreover, in another family, two siblings, a male and a female, carried two distinct mutations, respectively, a *MAPT* mutation Val75Ala and a *PSEN2* mutation Arg62Hys (already described in [69]).

As previously reported, in the Northern cohort, the frequency of individuals with *GRN* mutations was higher than those of other groups (115 [68%] of 169 individuals); whereas, individuals with *PSEN1* mutation were found more frequently in the Central (58 [44.6%] of 112) and in the Southern cohort (66 [35.5%] of 186). Interestingly, individuals with *APP* mutations share a similar and higher frequency in the Central and in the Southern cohorts (21 [18.8%] of 112 and 36 [19.4%] of 186, respectively) compared to the Northern cohort (1 [0.6%] of 169).

No significant differences in the number of men and women were shown among the genetic groups in the total Italian cohort. Regarding the age at onset in the different genetic groups, the lowest mean value was for the *PSEN1* gene (range, 23–73 years), with a significant decrease compared to *APP* (41–82 years), *PSEN2* (22–84 years), *GRN* (40–82 years)*,* and *C9orf72* (40–80 years) (*p* < 0.0001 for each comparison, one-way ANOVA test with Bonferroni post hoc correction) (Figure 2). *MAPT* was the second group with the lowest age at onset, with a significant decrease compared to *APP* (*p* = 0.0004), *PSEN2* (*p* = 0.0024), *GRN* (*p* < 0.0001), and *C9orf72* (*p* = 0.0049, one-way ANOVA test with Bonferroni post hoc correction). The only common mutation identified among the three different cohorts was Met146Leu, observed in *PSEN1* group. This mutation did not show a significant effect in terms of age at onset. Significant differences between the six genetic groups were also maintained, including sex, Italian origin (north, center, south), and family membership as covariate in the statistical model, used to evaluate the age at onset in the three cohorts. Interestingly, neither the origin nor the sex, but only the genetic group, were associated with age at onset. We confirmed a significant difference in age at onset between the *PSEN1* group and *APP* (*p* = 0.0002), *PSEN2* (*p* = 0.0005), *GRN* (*p* < 0.0001)*, C9orf72* (*p* = 0.0002), and between *MAPT* and *GRN* (*p* = 0.0009) (Linear Mixed Model adjusted for sex, origin, and family). We observed a similar range of age at onset (about 40 to 82) for *APP*, *GRN*, and *C9orf72* genetic group. The largest range of age at onset was observed in *PSEN2* genetic group (22–84), while *PSEN1*, *PSEN2*, and *MAPT* were the genetic groups with the youngest age at onset (22 and 23 years). 

Moreover, to identify the presence of variation in age at onset between mutations of the same gene, we selected the most represented mutations (*n* ≥ 5) for each genetic group. We found a significant difference in *APP* group, in particular an earlier age at onset for Val717Ile (*n* = 11, mean ± st. dev., 53.82 ± 5.72) compared to Ala713Thr (*n* = 25, 63.66 ± 11.19) (*p* = 0.0008, *t* test), and in the *PSEN1* group, in particular an earlier age at onset for Leu392Val (rs63751416, *n* = 10, 44 ± 12) compared to Cys92Ser (*n* = 7, 57.63 ± 4.66) (*p* = 0.0003) and to Ile143Val (rs63750322, *n* = 6, 58.17 ± 3.76) (*p* = 0.0002), and an earlier age at onset for Met146Leu (*n* = 45, 41.1 ±4.63) compared to Cys92Ser (*n* = 7, 57.63 ± 4.66) (*p* < 0.0001) and to Ile143Val (rs63750322, *n* = 6, 58.17 ± 3.76) (*p* < 0.0001) (one-way ANOVA test with Bonferroni post hoc correction). Significant differences between the mutations previously described in *APP* and *PSEN1* were also maintained, including sex and family membership as covariate in the statistical model, used to evaluate variation in age at onset between mutations in each gene (*APP*, Val717Ile vs. Ala713Thr, *p* = 0.0496; *PSEN1*, Leu392Val vs. Cys92Ser, *p* = 0.0004 and Leu392Val vs. Ile143Val, *p* = 0.0003; Met146Leu vs. Cys92Ser, *p* < 0.0001 and Met146Leu vs. Ile143Val, *p* < 0.0001) (Linear Mixed Model adjusted for sex and family). We did not include the Italian origin in the model due to the cohort peculiarity of several mutations. 

## 3. Materials and Methods

### 3.1. Study Design and Participants

In this study, we collected the data of AD/FTD patients and asymptomatic subjects (recruited over the last 30 years) belonging to pedigrees from the north, the center, and the south of Italy, and whose biological samples were already stored at the institutional biobank/biorepositories of three Italian centers that are part of the GARDENIA Consortium, in the context of the project GARDENIA “Genetic and epigenetic modulAtors in Rare neurodegenerative disease with DEmentia: a National study on autosomal dominant Alzheimer disease and genetic frontotemporal degeneration with dementia” funded by the European Union—Next Generation EU (*PNRR-MR1-2022-12375654*). The aim of the Consortium is to generate the first national collection of clinical and deep sequencing data on monogenic AD/FTD. We included participants carrying mutations in *APP*, *PSEN1*, *PSEN2*, *MAPT*, *GRN*, and *C9orf72* genes. Data were obtained from (i) Northern cohort: IRCCS Centro San Giovanni di Dio Fatebenefratelli BioBank Brescia, Italy (bbmri-eric ID: IT_138442378660827 and Orphanet Biobank) (*n* = 169 individuals); (ii) Central cohort: Azienda Ospedaliero Universitaria Careggi in Florence, Tuscany (*n* = 112); (iii) Southern cohort: Azienda Sanitaria Provinciale di Catanzaro, Calabria (*n* = 186). The clinical diagnosis of AD and FTD were made in accordance with international guidelines [5,35,70,71]. Data collected from the centers contain genetic group, individual mutation, sex, clinical diagnosis, and age at onset.

### 3.2. Statistical Analysis 

We categorized participants into the *APP*, *PSEN1*, *PSEN2*, *MAPT*, *GRN*, or *C9orf72* group according to the mutations present. We calculated the numbers and percentages of individuals within each genetic group by geographic location. We calculated the means and standard deviation for age at symptom onset in each genetic group and for the most represented mutations (defined as those with five or more carriers). The normality assumption of age at onset was assessed using the Shapiro–Wilk test. To examine the relationship between categorical variables and genetic groups, we employed the Chi-square test. We used the independent samples t-test or the ANOVA test followed by post hoc pairwise t-tests with Bonferroni adjustment to identify significant differences in age at onset among genetic groups or mutations. Subsequently, we applied a Linear Mixed Model to test differences in age at onset among genetic groups, adjusting for the fixed effects of sex and origin (north, center, or south Italy), and accounting for the random effects of subject code and family code to control for individual variability and familial clustering. We also used linear mixed effects modeling to test differences in age at onset among the most represented mutations within the same gene, adjusting for the fixed effect of sex and considering family code as a random effect. The origin was not included as a fixed effect factor, due to the cohort peculiarity of several mutations, resulting in a very high association between mutation and origin. Post hoc pairwise comparisons between genetic groups or mutations were conducted, and *p*-values were adjusted using Tukey’s method (the adjusted *p*-values are reported). All statistical tests were two-tailed, with statistical significance set at *p* < 0.05. These analyses were performed using Rstudio (R version: 4.3.2).

### 3.3. Ethics Committee

All participants provided written informed consent. The study protocol was approved by the local ethics committee (Prot. N. 63/2022; date of approval: 7 December 2022).

## 4. Conclusions

In this study, we aimed to complement previous regional phenotypic studies by conducting an Italian national study of age at symptom onset in individuals with mutations in AD/FTD related genes (i.e., *APP*, *PSEN1*, *PSEN2*, *MAPT*, *GRN*, and *C9orf72*). 

Italy is a country with a rich history of cultural influences: the Italian peninsula has been shaped by waves of conquest and settlement by different peoples for ages, and the country became unified only in the 19th century. Indeed, we observed a geographical variability in the frequency of mutations of AD/FTD genes. We showed that the most prevalent genetic group in the Northern cohort was the *GRN* one, due to the high number of individuals carrying the Leu271LeufsX10 mutation that was in fact first described in northern Italy, suggesting a founder effect from a common ancestor in the Middle Ages. The high number of Met146Leu carriers in *PSEN1*, a private mutation with a common ancestor in a Calabrian family dated around the year 1000, may explain the higher distribution of *PSEN1* genetic group in the Southern and Central cohorts. 

The analysis of phenotypic characteristics on the entire Italian cohort confirms previous studies regarding the high variability of age at onset across the genes implicated in AD and FTD. This variation is present not only at the gene level, but also between specific mutations in *APP* and *PSEN1* genes. We observed a similar range of age at onset (about 40 to 82) for *APP*, *GRN,* and the *C9orf72* genetic group. The largest range of age at onset was observed in the *PSEN2* genetic group (22–84), while *PSEN1*, *PSEN2*, and *MAPT* were the genetic groups with the youngest age at onset (22 and 23 years). Interestingly, neither the origin nor the sex, but only the genetic group, were associated with age at onset. 

Further study in identifying both the genetic and environmental factors that modify the phenotypes in all groups is needed. Our study reveals Italian regional differences among the most relevant AD/FTD causative genes, and emphasizes how the collaborative studies in rare diseases can provide new insights to expand the knowledge on the genetic/epigenetic modulators of age at onset.

## Figures and Tables

**Figure 1 ijms-25-07035-f001:**
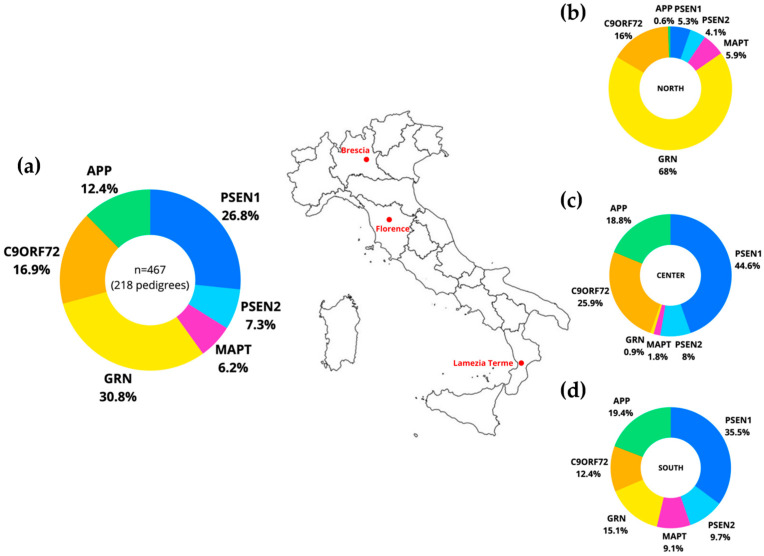
Frequency of each of the six genetic groups *APP*, *PSEN1*, *PSEN2*, *MAPT*, *GRN*, and *C9orf72* in the combined dataset of this study (**a**), in the Northern (**b**), in the Central (**c**), and in the Southern cohort (**d**) (*p* < 0.0001, Chi-square test).

**Figure 2 ijms-25-07035-f002:**
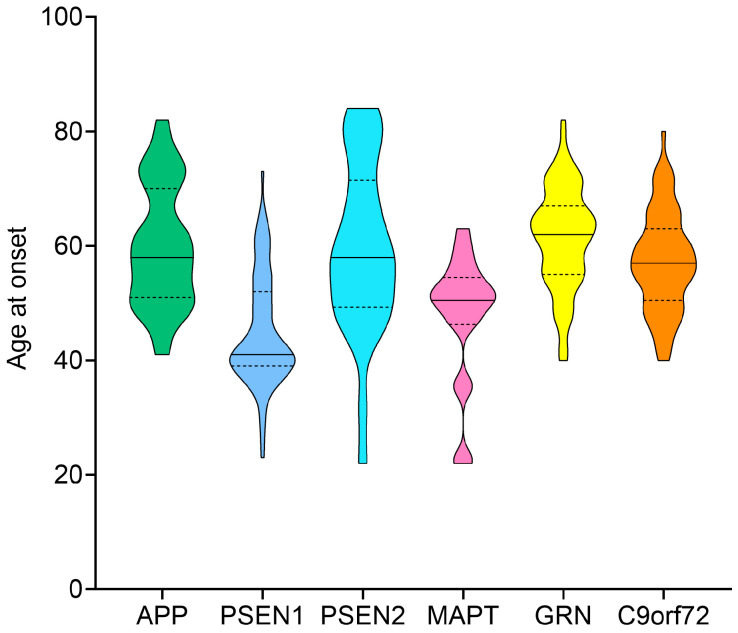
Violin plots of age at onset for the six genetic groups *APP*, *PSEN1*, *PSEN2*, *MAPT*, *GRN*, and *C9orf72* (*p* < 0.0001, Linear Mixed Model).

**Table 1 ijms-25-07035-t001:** Characteristics of the Italian cohort.

	*APP* (*n* = 58)	*PSEN1* (*n* = 125)	*PSEN2* (*n* = 34)	*MAPT* (*n* = 29)	*GRN* (*n* = 144)	*C9orf72* (*n* = 79)	*p*-Value
Number of families	21	32	19	16	79	57	
Sex (% female)	34.5	52.8	50.0	51.7	47.2	53.2	0.2676 ^a^
Age at onset	59.9 ± 10.7	44.9 ± 9.7	59.3 ± 15.2	48.0 ± 11.0	61.4 ± 8.9	57.4 ± 8.7	<0.0001 ^b^

*APP*, *APP* mutation carriers; *PSEN1*, *PSEN1* mutation carriers; *PSEN2*, *PSEN2* mutation carriers; *MAPT*, *MAPT* mutation carriers; *GRN*, *GRN* mutation carriers; *C9orf72*, *C9orf72* mutation carriers. ^a^ Chi-square; ^b^ One-way ANOVA test with Bonferroni post hoc correction. Means ± standard deviation.

## Data Availability

The data presented in this study are available in the Zenodo Data Repository at https://doi.org/10.5281/zenodo.11125535 [72].

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
