# Peer review of "Mutational Landscape of Alzheimer’s Disease and Frontotemporal Dementia: Regional Variances in Northern, Central, and Southern Italy"

_ijms, 2024, doi:10.3390/ijms25137035_

Round 1

Reviewer 1 Report

Comments and Suggestions for Authors

In their brief communication titled ” Mutational Landscape of Alzheimer’s Disease and Frontotemporal Dementia: Regional Variances in Northern, Central, and Southern Italy”, the authors study the frequency of early-onset Alzheimer’s disease (EOAD) and frontotemporal dementia  (FTD) gene variants in three regions of Italy.  Additionally, they study differences in the age-of-onset between disease-causing genes. Their cohort consists of 467 individuals from 218 pedigrees.  The authors report statistically and clinically significant variation in the age-of-onset (e.g. PSEN1 44.9+-9.7 vs. 59.9+-10.7 for APP).  They also find regional differences in the distribution of causative-genes across Italy: e.g. GRN is very common in Northern Italy, quite common in Southern Italy but rare in Central Italy.

Major comments:

1.      The distribution of causative genes behind EOAD/FTD is reported at the individual level and not at the family level. While this approach clarifies the absolute number of mutation carriers, it is prone to bias by family size. For example, as illustrated by Figure 1, PSEN1 individuals represent 44.6% of the cases in central Italy and C9orf72 carriers 25.9% of cases. However, at the family-level, PSEN1 mutations are found in 20/58 (34%) of families whereas C9orf72 is found in 27/58 (47%) of families. Reporting the data at the individual level is not wrong by any means, but I would add the family-level data and consider which is more informative to the reader.

2.      Adding to the previous comment, it would be informative to know from each of the three regions, what is the number/proportion of EOAD/FTD families with unknown genetic diagnosis – do the genes reported in this manuscript cover pretty much all the families or are we still missing a lot of causative genes?

3.      The age-of-onset comparison is done at the gene-level, which make sense. However, I’m wondering if there was any obvious variation between mutations of the same gene? Especially in GRN where 105/144 individuals had the same one mutation. If the word-limit prevents additional results, this could be shortly discussed in Conclusions.

4.      I would clarify what is the novelty of the study. In EOAD, it is well-established that PSEN1 has earliest onset followed by APP and PSEN2 (e.g. genereviews Alzheimer Disease Overview by Thomas D. Bird). Likewise, it is known that in FTD MAPT has the earliest age-of-onset followed by GRN and C9orf72 (e.g. Age at symptom onset and death and disease duration in genetic frontotemporal dementia: an international retrospective cohort study Moore et al Lancet Neurol 2020). It is also thoroughly reported that the age-of-onset can vary a lot even within families. The regional differences in causative gene frequency in Italy is novel but within the current scope of the manuscript the finding is mostly relevant within Italy.

Minor comments:

1.      Is the number of pedigrees (218) wrong in Figure 1? In Table 1, the sum of “Number of families” row is 224.

2.      It would be good to know what was the recruitment period of participants as the diagnostic criteria (and genetic tests available) might have changed considerably over time.

The study is well-written, in general well-made and has an important aim of elucidating region and gene-specific factors that modulate disease progression. However, the scope of the analyses is quite limited as is the novelty.

Reviewer 2 Report

Comments and Suggestions for Authors

General:

The study examines six genetic groups in three different cohorts in Italy. It shows that there are genetic and environmental factors that influence frontotemopral dementia. There was no difference between men and women. However, there was a difference in the age of onset of the disease

Minor concerns

(1) Subdivide the scaling of the y-axis in Figure 2 more strongly

(2) The IDs of the consortia are referred to in the material and methoid section. It would be good to write at least one sentence explaining the criteria here

(3) Informed consent and ethics are not explicitly mentioned. Please complete the information

(4) I am not a statistician but I usually see a more complex statistical analysis of such epidemiologic data sets. I would ask the authors to justify their statistics.

(5) Page 7, line 270: common ancestor: How far back is that?

(6) Page 7, line 279-281: Have any of these characteristics been described in another study? (I mean yes). If applicable, please refer.

Round 2

Reviewer 1 Report

Comments and Suggestions for Authors

The authors have adequately responded to all my comments. The addition of analyses on different variants of the same gene brings novelty to the letter and I find it suitable for publication.